# Molecular and Serological Detection of *Anaplasma phagocytophilum* in Dogs from Germany (2008–2020)

**DOI:** 10.3390/ani13040720

**Published:** 2023-02-17

**Authors:** Ingo Schäfer, Barbara Kohn, Cornelia Silaghi, Susanne Fischer, Cedric Marsboom, Guy Hendrickx, Elisabeth Müller

**Affiliations:** 1LABOKLIN GmbH and Co. KG., 97688 Bad Kissingen, Germany; 2Small Animal Clinic, Faculty of Veterinary Medicine, Freie Universität Berlin, 14163 Berlin, Germany; 3Institute of Infectology, Friedrich-Loeffler-Institute, 17493 Greifswald, Germany; 4Avia-GIS, 2980 Zoersel, Belgium

**Keywords:** Canine, ELISA, granulocytic anaplasmosis, IFAT, PCR, prevalence

## Abstract

**Simple Summary:**

The bacterium *Anaplasma phagocytophilum* can cause granulocytic anaplasmosis in domestic animals, wildlife, and humans. The pathogen is transmitted by ticks and is predominantly endemic in Central and Northern Europe. We discuss the percentages of dogs that tested positive for *A. phagocytophilum* by direct detection of the pathogen (4.9%) and by detecting antibodies (23.3%). We found a statistically significant impact of seasonality and years of testing on PCR results as well as sex, age, years of testing and seasonality on antibody results. The dynamics of infections with *A. phagocytophilum* in dogs in Germany are consistent with peaks in vector activity. There is a notable increase in canine granulocytic anaplasmosis in Germany over the time of this study, which calls for increased attention and demonstrates its rising importance in clinical practice.

**Abstract:**

*Anaplasma phagocytophilum* is an obligate intracellular bacterium that causes granulocytic anaplasmosis in domestic animals, wildlife, and humans and is primarily transmitted by ticks of the *Ixodes persulcatus* complex. This retrospective study aims to determine the percentages of dogs that tested positive for *A. phagocytophilum* in Germany. It included the results of direct (polymerase chain reaction [PCR]) and indirect (immunofluorescence antibody test [IFAT], antibody-enzyme-linked immunosorbent assay [ELISA]) detection methods performed in the laboratory LABOKLIN on canine samples provided by German veterinarians from 2008 to 2020. Out of a total of 27,368 dogs tested by PCR, 1332 (4.9%) tested positive, while 24,720 (27.4%) of the 90,376 dogs tested by IFAT/ELISA had positive serology. High rates of positive PCR results were observed in months with known peaks in vector activity, showing that the dynamics of *A. phagocytophilum* infections in dogs in Germany are consistent with vector activity. In dogs with a positive PCR result, peaks in serology could be observed four weeks after initial testing. Male and senior dogs had higher rates of positive serology. A possible impact of environmental factors such as changes in climate should be investigated further. Overall, the upward trend in positive test results over the years indicates that canine granulocytic anaplasmosis will continue to become increasingly important for veterinary medicine.

## 1. Introduction

*Anaplasma phagocytophilum* is a gram-negative obligate intracellular bacterium that causes granulocytic anaplasmosis in humans as well as in domestic and wild animals [1,2]. The distribution of *A. phagocytophilum* infections depends largely on vector occurrence and prevalence. Its primary vectors are ticks of the *Ixodes persulcatus* complex, including *Ixodes ricinus,* which can be found throughout Germany [3,4,5]. While much less frequent, blood transfusions are a second potential route of transmission in dogs [6,7] or humans [8].

*A. phagocytophilum* infection can be diagnosed by PCR testing or the detection of either antibodies (IFAT and/or ELISA) or morulae (microscopy of peripheral blood smears or buffy coats) [4]. A positive PCR result is indicative of acute infection [9,10], while a single positive antibody titer may be caused by past contact with the pathogen. Antibodies cannot be detected in the early stages of an acute infection prior to seroconversion [11]. A four-fold increase in antibody titer between two paired serum samples taken within four weeks is indicative of an acute infection with *A. phagocytophilum* [12]. Single positive antibody titers are of minimal diagnostic value but may be of epidemiological interest. Antibodies can persist for several months or even years following an active infection, pathogen contact, or resolution [13,14,15]. Cross-reactions with other Rickettsiales can further complicate the interpretation of serology results [4,16]. Considering these limitations of antibody testing, PCR is considered the gold standard for the diagnosis of acute infections [4]. The detection of morulae in neutrophilic granulocytes in peripheral blood smears is an alternative diagnostic method for acute infections only, but it is much less sensitive than PCR testing [6,17,18].

There are several previous reports on the rates of dogs in Germany with positive PCR or serology, with varying results (PCR: 4–6% [15,19,20], IFAT: 19–50% [15,19,21,22,23], ELISA: 19–21% [24,25]). The prevalence of *A. phagocytophilum* in dogs in Germany is largely unknown. Of the studies mentioned, only Krupka et al. [24] included all federal states in Germany. The aim of this study was to compare the frequency of dogs with positive serology and PCR in different federal states, the seasonal as well as annual distribution of positive test results, and the possible age and sex correlation of dogs that tested positive.

## 2. Materials and Methods

In this retrospective study, a total of 110,240 dogs were tested for *A. phagocytophilum* by direct and/or indirect detection methods between 2008 and 2020. Direct (PCR) and indirect (ELISA, IFAT) diagnostic assays for the detection of *A. phagocytophilum* in dogs were included. In cases with repeat samples for the same animal and/or detection method, only the results of either the first positive sample or the first sample overall were included. All diagnostic material used was surplus from samples provided to the commercial laboratory LABOKLIN (Bad Kissingen, Germany) by veterinarians in Germany between January 2008 and December 2020. According to the terms and conditions of the laboratory LABOKLIN as well as the decision of the government of Lower Franconia RUF-55.2.2.2532-1-86-5, no special permission from animal owners or the animal welfare commission was needed for additional testing on residual samples once diagnostics were complete.

The direct detection method used in this study was a qualitative TaqMan real-time PCR (Applied Biosystems/Life Technologies, target gene: 60-kDa heat shock protein [HSP60]) applied on 27,368 dogs out of EDTA or whole blood (*n* = 24,260), cerebrospinal fluid (*n* = 3034), synovial fluid (*n* = 67), or bone marrow samples (*n* = 7). Automated nucleic acid extraction was carried out on sample volumes of 200 μL using a commercially available kit (“MagNA Pure 96 DNA and Viral NA Small Volume Kit”, Roche Diagnostics GmbH, Mannheim, Germany) according to the manufacturer’s instructions. The resulting nucleic acid was eluted in a final volume of 100 μL. A TaqMan real-time PCR assay was performed to quantitatively examine for/the presence of *A. phagocytophilum*, targeting the 60-kDa heat shock protein (Gene Expression Assay, Applied Biosystems/Thermo Fisher Scientific, Waltham, MA, USA) on a LightCycler® 96 instrument (Roche Diagnostics GmbH, Mannheim, Germany). The reaction mixture was amplified as follows: denaturation at 95 °C for 30 s and 40 cycles at denaturation at 95 °C for five seconds each, and annealing/extension at 60 °C for 30 s. The PCR was applied as a qualitative assay (negative/positive). Ct-values below 35 were considered positive. Each PCR run included a negative and a positive control as well as an extraction control for each sample to check for nucleic acid extraction and PCR inhibition (“DNA Process Control Detection Kit”, Roche Diagnostics GmbH, Mannheim, Germany).

Serum samples of 90,367 dogs were tested for IgG antibodies by IFAT (MegaFLUO ANAPLASMA; MegaCor Diagnostik; >1:40 positive) and/or ELISA testing (*Anaplasma*-ELISA Dog; afosa GmbH; >11 LE positive), the latter having been added from 2017. The results of both methods were analyzed without discrimination due to their comparable sensitivity and specificity (sensitivity of 92.5% and specificity of up to 97% using 154 characterized sera from dogs) [26]. 

All dogs tested for *A. phagocytophilum* by PCR and/or serological testing were included in the study, independent of breed, age, or sex status. Dogs were subdivided by age and time of sample collection (by year and month). Descriptive statistical analysis was carried out using SPSS for Windows (version 28.0; International Business Machines Corporation). *p* < 0.05 was considered statistically significant. Data were checked for normal distribution by Kolmogorov–Smirnov testing and suitable tests were chosen for statistical analysis (Chi-square test, Mann–Whitney U test, Fisher’s exact test). Binomial logistic regression was performed to determine the effect of sex, age groups, and time of testing. The 95% CI for the proportion of dogs that tested positive by PCR and IFAT/ELISA was calculated by the Wilson procedure including correction for continuity.

## 3. Results

Of the 110,240 dogs tested for *A. phagocytophilum* by direct and/or indirect detection methods between 2008 and 2020, 25,724 tested positive by either or both methods: *A. phagocytophilum* was detected in 1332 of 27,368 (4.9%) dogs tested by PCR and 24,720 of 90,376 (27.4%) dogs tested by IFAT/ELISA. PCR testing was performed on whole blood/EDTA blood (1322 of 24,260 dogs tested positive; 5.4%), cerebrospinal fluid (10 of 3034 dogs tested positive, 0.3%), synovial fluid (67 dogs tested, no positive cases), and bone marrow (7 dogs tested, no positive cases). A total of 7504 dogs were tested by both PCR and antibody detection. Among these, 327 of 7504 (4.4%) dogs were double positive, while both tests were negative in 3427 of 7504 dogs (45.6%). Antibodies were detected in 3601 of 7504 dogs (48%) with a negative PCR result, while the remaining 149 of 7504 dogs (2%) had a positive PCR result but no detectable antibodies. 

Of the total 110,240 dogs in this study, the breed was known in 101,846 dogs (92.4%, most commonly mixed breeds [42,251/101,846 dogs, 41.5%]). The age was known in 96,723 dogs (87.7%, mean 5.9 years, median 6.0 years, range 0.2 to 19.0 years with a standard derivation of 3.83 years). Dogs were subdivided by age into five groups (Table 1). Results were not normally distributed between different age groups (Kolmogorov–Smirnov *p* < 0.001) and statistically significant changes were detected in both PCR (Mann –Whitney U test, *p* < 0.001, 154,666,055.500) and antibody testing (Mann–Whitney U test, *p* < 0.001, 825,081,293.000) (Table 1).

The sex of 101,505 out of 110,240 dogs was documented (92.1%, 52,237 out of 101,505 males [51.5%], 49,268 out of 101,505 females [48.5%]). There appeared to be no statistically significant difference in PCR results between the sexes (*χ^2^* = 4.665, *df* = 3, *p* = 0.196). Male dogs were however significantly more likely to have detectable antibodies than female dogs (Fisher’s exact test: *p* < 0.001).

The year of sample collection had a statistically significant impact on the results of both PCR and antibody testing (PCR: *χ^2^* = 25.264, *df* = 12, *p* = 0.014, φ = 0.030; antibody testing: *χ^2^* = 2845.303, *df* = 12, *p* < 0.001, φ = 0.177), as did the time of year (by month) a sample was taken (PCR: *χ^2^* = 775.053, *df* = 11, *p* < 0.001, φ = 0.168; antibody testing: *χ^2^* = 730.484, *df* = 11, *p* < 0.001, φ = 0.090) (Table 2).

More dogs tested positive in the summer (June to August, Figure 1), starting one month after the known peak activity for the potential vector *I. ricinus* in Germany [27]. There was a one-month lag between the peak of positive PCR results and that of antibody detection (Figure 1). 

Binomial logistic regression was carried out to determine the effect of sex, age groups, and time of sample collection (by year and calendar month) on the results of PCR and antibody testing (Table 3). Correlations between predictor variables were low (*r* < 0.70), indicating that multicollinearity was not a confounding factor in this analysis. Of the four variables entered into the regression model, two contributed significantly to positive results of PCR testing (years of testing *p* = 0.001, months of testing *p* < 0.001), while all four had an impact on antibody testing (sex, age groups, years, and months of testing: *p* < 0.001 respectively). Male dogs were more likely to have detectable antibodies (OR = 1.269, 95%-CI [1.228, 1.312]), while sex had no statistically significant effect on PCR results (*p* = 0.995). Older age (>7 years) also increased the rates of antibody detection (OR = 2.392, 95%-CI [2.314, 2.472]) but did not affect PCR results (*p* = 0.122). The odds of testing positive by either PCR (OR = 1.029, 95%-CI [1.011, 1.047]) or antibody testing (OR = 1.056, 95%-CI [1.051, 1.060]) increased over time (Table 3).

Location (as determined by the federal state in Germany) significantly impacted the rate of dogs with positive PCR results (*χ^2^* = 129.897; *df* = 12; *p* < 0.001; φ = 0.069) or detectable antibodies (*χ^2^* = 1053.471, *df* = 12; *p* < 0.001; φ = 0.108) to *A. phagocytophilum* (Table 4). The highest rates of dogs with positive PCR results were found in Berlin-Brandenburg (7.6%), Saxony (7.6%), and Thuringia (7.4%). The highest rates of dogs with detectable antibodies were seen in Bavaria (34.8%), Lower Saxony/Bremen (33.6%), and the Saarland (30.3%). A Appendix A demonstrates percentages of dogs that tested positive for *A. phagocytophilum* by PCR and antibody testing per year for all individual federal states [see Appendix A].

## 4. Discussion

To the best of the authors’ knowledge, this is the first study evaluating *A. phagocytophilum* detection by direct (PCR) and indirect (IFAT, ELISA) detection methods in dogs in Germany to span 13 years (2008–2020) and to include all federal states. Overall, 5.4% of dogs tested positive by PCR (on EDTA blood), and 27.4% of dogs had positive serology. These findings are consistent with previous studies reporting *A. phagocytophilum* detection rates of 4–6% by PCR [15,19,20] and antibody detection rates of 21–50% in individual federal states [15,19,21,22,23,24,25]. The range of *I. ricinus* appears to be expanding toward northern Europe, resulting in an increasing prevalence of *A. phagocytophilum* in these areas [28,29,30,31,32]. This is consistent with the apparent increase in affected dogs in the northeastern areas of Germany, as well as generally over time in our study. Fewer dogs had positive serology in southwestern federal states like Hesse (20%) and Baden Wuerttemberg (23.5%), as well as in Mecklenburg Western Pomerania (23.4%, Table 4). There is no centralized data on tick prevalence in Germany and so it was not currently possible to correlate the prevalence of *A. phagocytophilum* in ticks to these findings. Further research in this area would be of interest. 

Changes in climate, land use, wildlife reservoirs, and population density can affect the ranges and population sizes of many tick vectors like *I. ricinus*, which may have impacted the changes in *A. phagocytophilum* detection in dogs in Germany over time.

The findings from a limited number of cerebrospinal fluids, synovial fluids, and bone marrow samples tested by PCR support the current literature that peripheral EDTA blood may be superior for the detection of acute *A. phagocytophilum* infections [33].

Since 2015, the percentage of dogs with positive PCR results has remained between 5.0 and 5.7%. Between 2017 and 2020, IgG antibodies were consistently detected in over 30% of dogs (Table 2). From 2008 to 2020, there was an overall increase in *A. phagocytophilum* detection by PCR (OR = 1.029) or antibody detection by ELISA/IFAT (OR = 1.056) (Table 3). This trend might be at least partly due to increased awareness of *A. phagocytophilum* among owners and veterinarians. Between 2013 and 2016, a decrease in the rates of positive PCR or serology results could be observed in the south of Germany, possibly due to a localized drop in tick population or activity.

The prevalence of *A. phagocytophilum* infections detected by PCR and antibody testing exhibited statistically significant seasonality, with peaks in summer (*p* < 0.001 each, Table 3, Figure 1). Dogs tested in the summer months of June to August were more likely to test positive by PCR (OR = 3.233) and antibody testing (OR = 1.403), though to a lesser degree (Table 3). This is consistent with previous reports of increased clinical presentation and PCR detection between May and August [34,35,36,37]. 

Gethmann et al. analyzed abiotic factors and their influence on *I. ricinus* activity over a two-year period at several tick collection sites in Germany [27]. Month and season as well as ground and air temperature had a statistically significant impact on the number of ticks caught. The highest activity of *I. ricinus* ticks was seen between 20 and 23 °C for air temperature and 13 and 15 °C for ground temperature, with most ticks caught between April and July in Germany [27]. This is in accordance with the peaks of the percentages of dogs tested positive by PCR and antibody testing in summer in this study (Figure 1) and demonstrates a correlation between *I. ricinus* tick activity and molecular as well as serological detection of the pathogen *A. phagocytophilum* in dogs.

The slight delay between peaks in vector activity and seasonally increased pathogen detection by PCR may be due to the 1–2-week incubation period of *A. phagocytophilum* [6]. Antibody detection also appears to lag behind PCR results, at a rate that is consistent with the time needed for seroconversion. While *A. phagocytophilum* can be detected by PCR as soon as 2–3 days after infection [17,38], antibodies were not detectable until 10–22 days post-infection in experimental studies [6,17,38]. The majority of dogs tested by both methods had negative PCR results in the presence of detectable antibodies (45.6%), which could be interpreted as persistent antibody titers after previous pathogen contact. Pathogen detection by PCR without detectable antibodies was rare (2%) and is most likely due to an acute infection prior to seroconversion, possibly between 2–3 days and 10–22 days post-infection [6,17,38]. In 4.4% of dogs, both *A. phagocytophilum* PCR and serology were positive. These cases are most likely due to an acute infection with seroconversion or persistent antibodies from past contact with the pathogen with a superimposed acute infection. Even though cross-reactions in antibody testing, especially with other Rickettsiales, may occur and complicate the interpretation of serological results [4,16], our study demonstrated a consistent similarity between the percentages of dogs tested positive by PCR and antibody testing (Figure 1). This highlights the complementary of both diagnostic tests, and therefore their value in epidemiological studies.

Both sex and age had a statistically significant impact on the results of PCR and/or antibody testing. Male dogs were more likely to have detectable antibodies than female dogs (OR = 1.269), indicating more frequent pathogen contact and/or higher susceptibility in male than female dogs. In this regard, sex-dependent behavior patterns that could result in a higher risk of vector contact in males may merit further discussion. Previous studies have found no effect of sex or age on dogs that tested positive but did report that older dogs were more frequently seropositive, possibly due to an increased risk of exposure to ticks and *A. phagocytophilum* over time or the prolonged persistence of antibodies after pathogen contact [15,39,40]. This is consistent with our finding of higher rates of antibody detection in older dogs (OR = 2.392). Similarly, the seroprevalence in dogs aged up to 4 years (51.3–55.6%) in an Austrian study was less than that of older age groups (59.1–69.2%) [41]. These effects could not be shown in the binominal logistic regression for PCR results (Table 3). In this study, dogs older than 10 years were significantly more likely to test positive by either detection method (Table 1). Age-related immunosuppression and comorbidities may be possible explanations for these trends, especially considering similar reports regarding *A. phagocytophilum* infections in humans older than 50 years [42].

### Limitations of the Study

The limitations of this study include its retrospective design. Potentially important background history was unavailable, including ectoparasite prophylaxis, living conditions, travel history, or the indication for investigations (e.g., screening, clinically sick dogs suspicious for granulocytic anaplasmosis). Evaluation of the breeds of dogs tested for *A. phagocytophilum* by direct and indirect detection methods might be of interest, as, for example, hunting dogs or dogs living in rural areas may have a higher risk for tick infestation and therefore for pathogen contact and/or infection. These factors could all impact the rates of *A. phagocytophilum* in the dogs tested, though their effect is likely minimized by the large number of dogs included in the study. Antibody testing is complicated by potential cross-reactions with other Rickettsiales. In Europe, *Anaplasma platys* or *Ehrlichia canis* would pose the greatest risk, though no Rickettsiales are currently endemic in Germany [43]. Possible infections would therefore most likely be linked to stays abroad, which could not be assessed. Dogs were also not tested for possible coinfections with other pathogens. 

## 5. Conclusions

This is the first nationwide study in Germany to report the detection rates of *A. phagocytophilum* in dogs by PCR and antibody testing over a period of 13 years. Both pathogen detection by PCR and antibody detection in dogs in Germany show peaks that correlate to vector activity, especially in the summer months. The overall upward trend of positive test results over the years may indicate that canine granulocytic anaplasmosis will continue to become increasingly important for veterinary medicine in Germany. Sex and age appear to have little or no effect on PCR results. Older dogs seem to be more likely to be seropositive. The rates of positive PCR or serology results differ between federal states in Germany. More research may be necessary into possible causes, such as local climate or the prevalence of *A. phagocytophilum* in local tick populations. Ectoparasite prophylaxis in dogs to avoid the development of reservoirs, as well as the screening of potential blood donors is highly recommended. 

## Figures and Tables

**Figure 1 animals-13-00720-f001:**
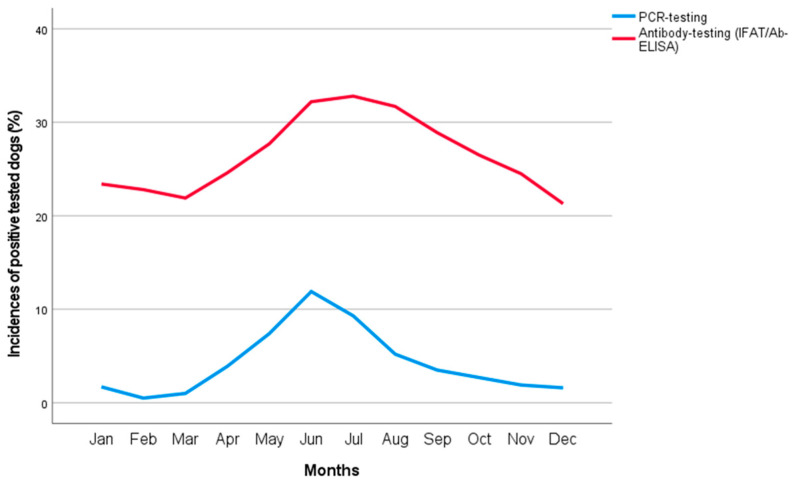
Monthly distribution of dogs that tested positive for *Anaplasma phagocytophilum* by direct (PCR; blue line) and indirect detection methods (IFAT, ELISA; red line) from 2008 to 2020.

**Table 1 animals-13-00720-t001:** Number of positive tests in dogs including direct and indirect detection methods for *Anaplasma phagocytophilum* in dogs living in Germany from 2008–2020 sorted by age (*n* positive/*N* total (% [95% CI lower limit; 95% CI upper limit]).

Age-Group	PCR	IFAT/ELISA
0–2 years (junior)	206/4934 (4.2 [3.7; 4.8])	2557/20,749 (12.3 [11.9; 12.8])
>2–7 years (adult)	478/9529 (5.0 [4.6; 5.5])	8924/31,198 (28.6 [28.1; 29.1])
>7–10 years (mature)	285/6019 (4.7 [4.2; 5.3])	6473/16,349 (39.6 [38.8; 40.4])
>10–13 years (senior)	213/3809 (5.6 [4.9; 6.4])	3704/8757 (42.3 [41.3; 43.3])
>13 years (geriatric)	49/762 (6.4 [4.8; 8.5])	600/1475 (40.7 [38.2; 43.2])
Total	1231/25,053 (4.9 [4.7; 5.2])	22,258/78,528 (28.3 [28.0; 28.7])
Mann–Whitney U test	*p* < 0.001	*p* < 0.001

CI: confidence interval; ELISA: Antibody-enzyme-linked immunosorbent assay; IFAT: immunofluorescence antibody test; PCR: Polymerase chain reaction; PCR: Mann–Whitney U test, *p* < 0.001, 154,666,055.500; IFAT/ELISA: Mann–Whitney U test, *p* < 0.001, 825,081,293.000.

**Table 2 animals-13-00720-t002:** Number of positive tests in dogs including direct and indirect detection methods for *Anaplasma phagocytophilum* in dogs living in Germany from 2008–2020 sorted by years of testing (*n* positive/*N* total (% [95% CI lower limit; 95% CI upper limit]).

Year	PCR	IFAT/ELISA	IFAT	ELISA
2008	40/850 (4.7 [3.4; 6.4])	531/2988 (17.8 [16.4; 19.2])	531/2988 (17.8 [16.4; 19.2])	
2009	53/1194 (4.4 [3.4; 5.8])	1054/4082 (25.8 [24.5; 27.2])	1054/4082 (25.8 [24.5; 27.2])	
2010	42/1245 (3.4 [2.5; 4.6])	1187/4160 (28.5 [27.2; 29.9])	1187/4160 (28.5 [27.2; 29.9])	
2011	55/1182 (4.7 [3.6; 6.1])	1491/5174 (28.8 [27.6; 30.1])	1491/5174 (28.8 [27.6; 30.1])	
2012	58/1355 (4.3 [3.3; 5.5])	1669/6187 (27.0 [25.9; 28.1])	1669/6187 (27.0 [25.9; 28.1])	
2013	46/1461 (3.1 [2.3; 4.2])	1698/6539 (26.0 [24.9; 27.1])	1698/6539 (26.0 [24.9; 27.1])	
2014	84/1907 (4.4 [3.5; 5.4])	1534/7875 (19.5 [18.6; 20.4])	1534/7875 (19.5 [18.6; 20.4])	
2015	100/1887 (5.3 [4.4; 6.4])	984/7616 (12.9 [12.2; 13.7])	984/7616 (12.9 [12.2; 13.7])	
2016	124/2378 (5.2 [4.4; 6.2])	1208/7487 (16.1 [15.3; 17.0])	1208/7487 (16.1 [15.3; 17.0])	
2017	167/2955 (5.7 [4.9; 6.6])	3383/9146 (37.0 [36.0; 38.0])	451/1365 (33.0 [30.6; 35.6])	3206/8949 (35.8 [34.8; 36.8])
2018	157/3160 (5.0 [4.3; 5.8])	3183/8892 (35.8 [34.8; 36.8])	305/1085 (28.1 [25.5; 30.9])	3032/8729 (34.7 [33.7; 35.7])
2019	190/3598 (5.3 [4.6; 6.1])	2998/9390 (31.9 [31.0; 32.9])	248/964 (25.7 [23.0; 28.6])	2848/9235 (30.8 [29.9; 31.8])
2020	216/4196 (5.1 [4.5; 5.9])	3800/10,840 (35.1 [34.2; 36.0])	254/1092 (23.3 [20.8; 25.9])	3639/10,668 (34.1 [33.2; 35.0])
Total	1332/27,368 (4.9 [4.6; 5.1])	24,720/90,376 (27.4 [27.1; 27.6])	12,614/56,614 (22.3 [21.9; 22.6])	12,725/37,581 (33.9 [33.4; 34.3])

CI: confidence interval; ELISA: Antibody-enzyme-linked immunosorbent assay; IFAT: Immunofluorescence antibody test; PCR: Polymerase chain reaction; Total: *χ^2^* = 2370.973; *df* = 12; *p* < 0.001; φ = 0.147; PCR: *χ^2^* = 25.264; *df* = 12; *p* = 0.014; φ = 0.03; IFAT/ELISA: *χ^2^* = 2845.303, *df* = 12; *p* < 0.001; φ = 0.177; IFAT: *χ^2^* = 1120.448, *df* = 12; *p* < 0.001; φ = 0.141; ELISA: *χ^2^* = 56.346, *df* = 3; *p* < 0.001; φ = 0.039.

**Table 3 animals-13-00720-t003:** Binominal logistic regression analysis in dogs tested for *Anaplasma phagocytophilum* by direct (PCR) and indirect detection methods (IFAT, ELISA) in 74973 dogs with known sex, age, years of testing, and months of testing from 2008 to 2020.

	B	SE	Wald	*p*	Odds Ratio	95%-CI for Odds Ratio	
						Lower Bound	Upper Bound
PCR testing							
Sex	0.000	0.060	0.000	0.995	1.000	0.888	1.125
Age	0.094	0.061	2.388	0.122	1.099	0.975	1.238
Years	0.029	0.009	10.582	0.001	1.029	1.011	1.047
Season	1.173	0.061	370.629	<0.001	3.233	2.869	3.643
Constant	−61.158	17.721	11.910	<0.001			
Antibody testing (IFAT/ELISA)							
Sex	0.238	0.017	203.113	<0.001	1.269	1.228	1.312
Age	0.872	0.017	2656.929	<0.001	2.392	2.314	2.472
Years	0.054	0.002	623.234	<0.001	1.056	1.051	1.060
Season	0.338	0.017	379.519	<0.001	1.403	1.356	1.451
Constant	−117.757	4.868	585.175	<0.001			

B: unstandardized regression weight; CI: confidence interval; ELISA: Antibody-enzyme-linked immunosorbent assay; IFAT: Immunofluorescence antibody test; PCR: Polymerase chain reaction; SE: standard deviation to the mean; Wald: Wald chi-squared test; Variables entered: sex: male, age: > 7 years, year, season: summer; Degrees of freedom were 1 for all Wald statistics.

**Table 4 animals-13-00720-t004:** Number of positive tests in dogs including direct and indirect detection methods of *Anaplasma phagocytophilum* in dogs living in Germany sorted by federal states of the submitting veterinarians (*n* positive/*N* total (% [95% CI lower limit; 95% CI upper limit]).

Federal State	PCR	IFAT/ELISA
Baden-Wuerttemberg	118/2887 (4.1 [3.4; 4.9])	2111/8968 (23.5 [22.7; 24.4])
Bavaria	139/2721 (5.1 [4.3; 6.0])	2511/7215 (34.8 [33.7; 35.9])
Berlin-Brandenburg	231/3038 (7.6 [6.7; 8.6])	1062/3475 (30.6 [29.0; 32.1])
Hesse	74/2496 (3.0 [2.4; 3.7])	1943/9739 (20.0 [19.2; 20.8])
Lower Saxony/Bremen	247/4341 (5.7 [5.0; 6.4])	5733/17,056 (33.6 [32.9;34.3])
Mecklenburg Western Pomerania	28/419 (6.7 [4.6; 9.6])	158/676 (23.4 [20.3; 26.8])
North Rhine Westphalia	237/6124 (3.9 [3.4; 4.4])	6913/26,786 (25.8 [25.3;26.3])
Rhineland Palatinate	55/1734 (3.2 [2.4; 4.1])	1358/6286 (21.6 [20.6; 22.6])
Saarland	28/748 (3.7 [2.5; 5.4])	672/2216 (30.3 [28.4; 32.3])
Saxony	76/1006 (7.6 [6.0; 9.4])	471/1817 (25.9 [23.9; 28.0])
Saxony-Anhalt	16/383 (4.2 [2.5; 6.8])	273/930 (29.4 [26.5; 32.4])
Schleswig-Holstein/Hamburg	52/1051 (4.9 [3.8; 6.5])	1176/4046 (29.1 [27.7; 30.5])
Thuringia	31/420 (7.4 [5.2; 10.4])	339/1166 (29.1 [26.5; 31.8])
Total	1332/27,368 (4.9 [4.6; 5.1])	24,720/90,376 (27.4 [27.1; 27.6])
Chi-square test	*p* < 0.001	*p* < 0.001

CI: confidence interval; ELISA: Antibody-enzyme-linked immunosorbent assay; IFAT: Immunofluorescence antibody test; PCR: Polymerase chain reaction; Total: *χ^2^* = 1048.245; *df* = 12; *p* < 0.001; φ = 0.098); PCR: *χ^2^* = 129.897; *df* = 12; *p* < 0.001; φ = 0.069); IFAT/ELISA: *χ^2^* = 1053.471, *df* = 12; *p* < 0.001; φ = 0.108).

## Data Availability

Not applicable.

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
