# Peer review of "Molecular and Serological Detection of Anaplasma phagocytophilum in Dogs from Germany (2008–2020)"

_animals, 2023, doi:10.3390/ani13040720_

Round 1

Reviewer 1 Report

The study has a lot of limitations, and even the authors indicate on many of tem. 

One of the aim of te study was to identify possible risk factors for AP infection. In fact only sex, age,  and origin of the animals wer studied. Sex had no statistically significant  effect on PCR results, and old age (> 7 years) increased the odds of a positive anti body test (this is common knowledge and doesn't need to be study). Real risk factors are: ectoparasite prophylaxis, living conditions,  contact wit other animals - but they were not studied.

We know very little about conditions of PCR. Were te primers used in real-time PCR  designed specific for A.phagocytopilum species?

There is a great diversity of AP genotypes among different host species based on several genetic markers. The authors seemed to be satisfied with just identification of the Anaplasma species and did not attempt any further genotyping. I would encourage the authors to make such attempts as detection of AP infection in dogs is nothing new and should be expanded. There are growing numbers of publications on AP diversity, including key papers by Jahfari S. et al. 2014, Dugat T. et al. 2014, 2016 and most recent Lagenwalder DB et al. 2020 (on the use of different genetic markers, i.e. ankA, msp4, groEl). Below is information from the Matei I. et al. review 2019 regarding the classification of genotypes/ecotypes and their host specificity. These issues could be discussed in the present study.

From Matei I. et al. review 2019 [Parasites and Vectors]: ‘Different authors published studies of genetic variants using different terminology, such as ecotype (groEL), cluster (ankA) or genotype (msp4) [16, 20, 21]. “Ecotype” refers to the hosts specificity of certain genotypes; “cluster” involves a deeper phylogenetic approach, while “genotype” is based on a purely genetic analysis. To refer to all mentioned terms, “genetic group” is used here. Different correlations of genetic variants have been found amongst vertebrate hosts, tick vectors and geographical locations. Infected humans, whether from Europe or America seem to share related strains belonging to the same genetic group [16, 20, 21]. Domestic animals like horses, dogs and cats, wild animals like red deer (Cervus elaphus), wild boars (Sus scrofa), red foxes (Vulpes vulpes) and hedgehogs (Erinaceus spp.) are harbouring strains with zoonotic potential related with human strains, while roe deer (Capreolus capreolus), rodents and birds seem to carry genetically distant strains [16, 19, 20, 21].

So with the limited number of positive PCR samples  typing would help get novel results.

The paper is more suitable for publication in local German journals like:BMTW or Tierarztliche Praxis.

In a present form I do not recommend to publish the article in Animals

Author Response

16-January-2023

Dear reviewer #1,

We appreciate the time and effort that you have dedicated to providing valuable feedback on our manuscript. We are grateful for your insightful comments, which have improved the paper. We have incorporated changes that reflect all your comments/suggestions. All changes are highlighted in red in the revised manuscript. Please see below our point-by-point response. If any responses may remain unclear or if you wish additional changes, please do not hesitate to let us know.

Reviewer #1

One of the aim of the study was to identify possible risk factors for AP infection. In fact only sex, age, and origin of the animals were studied. Sex had no statistically significant effect on PCR results, and old age (> 7 years) increased the odds of a positive anti body test (this is common knowledge and doesn't need to be study). Real risk factors are: ectoparasite prophylaxis, living conditions, contact with other animals - but they were not studied.

We agree with the reviewer’s opinion, that including the data regarding ectoparasite prophylaxis, living conditions and tick contact are of interest. However, we were not able to collect this information as a commercial diagnostic laboratory. We indicated this in the limitations-section in our manuscript (line 296 ff).

We know very little about conditions of PCR. Were the primers used in real-time PCR designed specific for A. phagocytophilum species?

We have added a more detailed description of the PCR used in our study in the manuscript.

Line 82 ff: “The direct detection method used in this study was a qualitative TaqMan real-time PCR (Applied Biosystems/Life Technologies, target gene: 60-kDa heat shock protein [HSP60]) on EDTA blood, whole blood, cerebrospinal fluid, synovial fluid, or bone marrow samples. Automated nucleic acid extraction was carried out on sample volumes of 200 μl using a commercially available kit ("MagNA Pure 96 DNA and Viral NA Small Volume Kit", Roche Diagnostics GmbH, Mannheim, Germany) according to manufacturer instructions. The resulting nucleic acid was eluted in a final volume of 100 μl. A TaqMan re-al-time PCR assay was performed to quantitatively examine for/the presence of Anaplasma phagocytophilum, targeting the 60-kDa heat shock protein (Gene Expression Assay, Applied Biosystems/Thermo Fisher Scientific, Waltham, USA) on a LightCycler® 96 instrument (Roche Diagnostics GmbH, Mannheim, Germany). Reaction mixture was amplified as follows: denaturation at 95°C for 30 seconds and 40 cycles at denaturation at 95°C for five seconds each, annealing/extension at 60°C for 30 seconds. The PCR was applied as qualitative assay (negative/positive). Ct-values below 35 were considered positive. Each PCR run included a negative and a positive control as well as an extraction control for each sample to check for nucleic acid extraction and PCR inhibition (“DNA Process Control Detection Kit”, Roche Diagnostics GmbH, Mannheim, Germany).”

There is a great diversity of AP genotypes among different host species based on several genetic markers. The authors seemed to be satisfied with just identification of the Anaplasma species and did not attempt any further genotyping. I would encourage the authors to make such attempts as detection of AP infection in dogs is nothing new and should be expanded. There are growing numbers of publications on AP diversity, including key papers by Jahfari S. et al. 2014, Dugat T. et al. 2014, 2016 and most recent Lagenwalder DB et al. 2020 (on the use of different genetic markers, i.e. ankA, msp4, groEl). Below is information from the Matei I. et al. review 2019 regarding the classification of genotypes/ecotypes and their host specificity. These issues could be discussed in the present study. From Matei I. et al. review 2019 [Parasites and Vectors]: ‘Different authors published studies of genetic variants using different terminology, such as ecotype (groEL), cluster (ankA) or genotype (msp4) [16, 20, 21]. “Ecotype” refers to the hosts specificity of certain genotypes; “cluster” involves a deeper phylogenetic approach, while “genotype” is based on a purely genetic analysis. To refer to all mentioned terms, “genetic group” is used here. Different correlations of genetic variants have been found amongst vertebrate hosts, tick vectors and geographical locations. Infected humans, whether from Europe or America seem to share related strains belonging to the same genetic group [16, 20, 21]. Domestic animals like horses, dogs and cats, wild animals like red deer (Cervus elaphus), wild boars (Sus scrofa), red foxes (Vulpes vulpes) and hedgehogs (Erinaceus spp.) are harbouring strains with zoonotic potential related with human strains, while roe deer (Capreolus capreolus), rodents and birds seem to carry genetically distant strains [16, 19, 20, 21].

We agree to the reviewer that genotyping of A. phagocytophilum is of interest and may provide interesting information about the genotypes being present in Germany. However, this was not the focus of our study. We focused on the trend of A. phagocytophilum infections in dogs in Germany including a time frame of 13 years, detection rates of PCR and serology, as well as potential impact of tick activity.

So with the limited number of positive PCR samples typing would help get novel results.

We agree to the reviewer. Please see above.

The paper is more suitable for publication in local German journals like: BMTW or Tierarztliche Praxis.

We believe that this study and its findings might be of great interest for the readership of “Animals”, as it covers additional aspects apart from epidemiological survey in Germany, e.g. risk factors.

Reviewer 2 Report

Dear author, I hope I can collaborate with the improvement of your manuscript. Here are some comments:

In general, the text needs language revision.

Line 18: The abbreviation of the genus name in parentheses is not necessary. Taxonomic nomenclature norms are clear about this. Remove "(A.)".

Line 19: Remove the comma after "wildlife".

Line 27: The abbreviation of the genus name in parentheses is not necessary. Taxonomic nomenclature norms are clear about this. Remove "(A.)".

Line 28: Remove the comma after "wildlife".

Line 28: Ixodes persulcatus must be in italic.

Line 30: A. phagocytophilum must be in italic.

Line 31: Replace "Polymerase" with "polymerase".

Line 34-35: The presentation of the data is confusing.

Line 35-36: The phrase "A higher percentage of PCR positive cases correlated with high vector activity" is not understandable. Higher than what?

Put the Keywords in alphabetical order.

Line 45: The abbreviation of the genus name in parentheses is not necessary. Taxonomic nomenclature norms are clear about this. Remove "(A.)".

Line 45: Replace "Gram-negative" with "gram-negative".

Line 45: Remove comma after "gram-negative".

Line 47: The abbreviation of the genus name in parentheses is not necessary. Taxonomic nomenclature norms are clear about this. Remove "(I.)".

Line 48: Replace "I. ricinus" with "Ixodes ricinus" as taxonomic norms.

Line 48: Replace "Ixodes ricinus" with "I. ricinus".

Line 49: In place of "mainly", "especially" might be better.

Line 50-51: The sentence "While much less frequent, blood transfusions are a second potential route of transmission between infected dogs [6,7] and infected humans [8]." can be understood there is transfusion between dogs and humans.

Line 64: Replace "may be" with "maybe".

Line 67-70: This paragraph should belong to the discussion and not the introduction. 

Line 85-86: Remove"from 2008 to 2020". This was mentioned earlier.

Mentions of Tables and Figures in Materials and Methods are meaningless. Do them in the results.

Line 117: Replace "19.0 years, standard derivation 3.83 years" with "19.0 years and standard derivation 3.83 years".

Tables must not have vertical lines. See example at: https://www.mdpi.com/2076-2615/13/2/252

In tables, not all thousands have commas and some percentages have commas instead of dots. Review the presentation of numerical data.

Line 152-154: This should be up for discussion.

Figure 2 should also be in the discussion.

Line 172-173: Replace "sex, age groups, years of testing, months of testing" with "sex, age groups, years of testing and months of testing".

Line 197: additional material is in another language.

Line 211-214: This information is very long and does not say much. Summarize to a maximum of two lines.

Line 215-218: What is the importance of comparing occurrence/prevalence values? Occurrence/prevalence values will always vary, according to the location, type and size of the sample, period, etc. The authors' own statement is strange "and antibody detection rates of 21-50%". How can numerical data look like 20% and 50% at the same time? The authors can even describe that during the period xx to the period xx, other isolated studies demonstrate a variation of xx% to xx%, but they cannot infer that they are data that are similar, unless a compatible statistical comparison is made. The remainder of the paragraph makes sense, as the authors compare the data from their study, comparing areas with each other, pointing out causes for the highest and lowest occurrences. This is valid, but the statement in these lines (215-218) is not reasonable.

Line 227-230: There are no references mentioned here.

Line 235-246: There are no references mentioned here. A lot of information is mentioned here, and without references it is just scientific speculation.

Line 270: Replace "with the pathogen with a superimposed acute infection" with "with the pathogen and a superimposed acute infection".

Line 310: Replace "Sex and age play have little or no effect on PCR results" with "Sex and age play little or no effect on PCR results".

Lines 324, 325, 326 and 335: Periods are missing at the end of sentences.

Author Response

Answer to the comments of reviewer #2

16-January-2023

Dear reviewer #2,

We appreciate the time and effort that you have dedicated to providing valuable feedback on our manuscript. We are grateful for your insightful comments, which have improved the paper. We have incorporated changes that reflect all your comments/suggestions. All changes are highlighted in red in the revised manuscript. Please see below our point-by-point response. If any responses may remain unclear or if you wish additional changes, please do not hesitate to let us know.

Reviewer #2

Dear author, I hope I can collaborate with the improvement of your manuscript. Here are some comments:

In general, the text needs language revision.

The manuscript was reviewed again by a native English-speaking colleague.

Line 18: The abbreviation of the genus name in parentheses is not necessary. Taxonomic nomenclature norms are clear about this. Remove "(A.)".

Has been done.

Line 19: Remove the comma after "wildlife".

Has been changed.

Line 27: The abbreviation of the genus name in parentheses is not necessary. Taxonomic nomenclature norms are clear about this. Remove "(A.)".

Has been done.

Line 28: Remove the comma after "wildlife".

Has been changed.

Line 28: Ixodes persulcatus must be in italic.

Has been done.

Line 30: A. phagocytophilum must be in italic.

Has been done.

Line 31: Replace "Polymerase" with "polymerase".

Has been done.

Line 34-35: The presentation of the data is confusing.

Has been changed.

Line 34 f: “Out of a total of 27,368 dogs tested by PCR, 1,332 (4.9%) tested positive, while 24,720 (27.4%) of the 90,376 dogs tested by IFAT/ELISA had positive serology.”

Line 35-36: The phrase "A higher percentage of PCR positive cases correlated with high vector activity" is not understandable. Higher than what?

Has been changed.

Line 36 ff: “High rates of positive PCR results were observed in months with known peaks in vector activity, showing that the dynamics of A. phagocytophilum infections in dogs in Germany are consistent with vector activity.”

Put the Keywords in alphabetical order.

Has been changed.

Line 43: “Canine, ELISA, granulocytic anaplasmosis, IFAT, PCR, prevalence”

Line 45: The abbreviation of the genus name in parentheses is not necessary. Taxonomic nomenclature norms are clear about this. Remove "(A.)".

Has been done.

Line 45: Replace "Gram-negative" with "gram-negative".

Has been done.

Line 45: Remove comma after "gram-negative".

Has been done.

Line 47: The abbreviation of the genus name in parentheses is not necessary. Taxonomic nomenclature norms are clear about this. Remove "(I.)".

Has been done.

Line 48: Replace "I. ricinus" with "Ixodes ricinus" as taxonomic norms.

Has been done.

Line 48: Replace "Ixodes ricinus" with "I. ricinus".

The authors prefer writing out the species here as it is placed in the beginning of the sentence.

Line 49: In place of "mainly", "especially" might be better.

Has been changed.

Line 48 f: “The distribution of A. phagocytophilum infections depends largely on vector occurrence and prevalence”

Line 50-51: The sentence "While much less frequent, blood transfusions are a second potential route of transmission between infected dogs [6,7] and infected humans [8]." can be understood there is transfusion between dogs and humans.

Has been changed.

Line 50 f: “While much less frequent, blood transfusions are a second potential route of transmission in dogs [6,7] or humans [8].”

Line 64: Replace "may be" with "maybe".

Has been changed.

Line 64 ff: “The detection of morulae in neutrophilic granulocytes in peripheral blood smears is an alternative diagnostic method for acute infections only, but it is much less sensitive than PCR testing [6,17,18].”

Line 67-70: This paragraph should belong to the discussion and not the introduction.

We agree to this comment and revised this section of the manuscript.

Line 68 ff: “There are several previous reports on the rates of dogs in Germany with positive PCR or serology, with varying results (PCR: 4 - 6 % [15,19,20], IFAT: 19 – 50 % [15,19,21-23], ELISA: 19 - 21 % [24,25]). The prevalence of A. phagocytophilum in dogs in Germany is largely unknown. Of the studies mentioned previously, only Krupka et al. [24] included all federal states in Germany.”

Line 85-86: Remove "from 2008 to 2020". This was mentioned earlier.

Has been changed.

Mentions of Tables and Figures in Materials and Methods are meaningless. Do them in the results.

The whole “Material and Methods-section” of the manuscript was revised according to the reviewer’s suggestion.

Line 117: Replace "19.0 years, standard derivation 3.83 years" with "19.0 years and standard derivation 3.83 years".

Has been changed.

Line 131 ff: “The age was known in 96,723/110,240 dogs (87.7%, mean 5.9 years, median 6.0 years, range 0.2 to 19.0 years with a standard derivation of 3.83 years).”

Tables must not have vertical lines. See example at: https://www.mdpi.com/2076-2615/13/2/252. In tables, not all thousands have commas and some percentages have commas instead of dots. Review the presentation of numerical data.

We carefully revised all tables included in the manuscript according to the reviewer’s comments.

Line 152-154: This should be up for discussion.

In our opinion, this should be stated in the results section, as it is description of data and does not include any discussion of results.

Figure 2 should also be in the discussion.

Has been done.

Line 172-173: Replace "sex, age groups, years of testing, months of testing" with "sex, age groups, years of testing and months of testing".

Has been done.

Line 181 ff: “Of the four variables entered the regression model, two contributed significantly to positive results of PCR testing (years of testing P = 0.001, months of testing P < 0.001), while all four had an impact on antibody testing (sex, age groups, years, and months of testing: P < 0.001 respectively).”

Line 197: additional material is in another language.

The term “additional file” was replaced by “supplementary file”. The file itself was reviewed for language purpose.

Line 211-214: This information is very long and does not say much. Summarize to a maximum of two lines.

We recommend not performing any changes here as this statement highlights the updated information for research purpose of our results.

Line 215-218: What is the importance of comparing occurrence/prevalence values? Occurrence/prevalence values will always vary, according to the location, type and size of the sample, period, etc. The authors' own statement is strange "and antibody detection rates of 21-50%". How can numerical data look like 20% and 50% at the same time? The authors can even describe that during the period xx to the period xx, other isolated studies demonstrate a variation of xx% to xx%, but they cannot infer that they are data that are similar, unless a compatible statistical comparison is made. The remainder of the paragraph makes sense, as the authors compare the data from their study, comparing areas with each other, pointing out causes for the highest and lowest occurrences. This is valid, but the statement in these lines (215-218) is not reasonable.

In our opinion, the data is presented and discussed in a reasonable way in this section of the manuscript.

Line 227-230: There are no references mentioned here.

As it is a more general and obvious statement, we do not see the need to include a reference here.

Line 235-246: There are no references mentioned here. A lot of information is mentioned here, and without references it is just scientific speculation.

This is discussion of our data without any links to references. May you please specify your comment?

Line 270: Replace "with the pathogen with a superimposed acute infection" with "with the pathogen and a superimposed acute infection".

Has been changed.

Line 276 ff: “These cases are most likely due to an acute infection with seroconversion or persistent antibodies from past contact with the pathogen with a superimposed acute infection.”

Line 310: Replace "Sex and age play have little or no effect on PCR results" with "Sex and age play little or no effect on PCR results".

This section was revised language wise and intense changed were performed (line 279 ff).

Lines 324, 325, 326 and 335: Periods are missing at the end of sentences.

Has been added.

Reviewer 3 Report

This is interesting and needed epidemiological analysis of canine granulocytic anaplasmosis in Germany. In my opinion the study lacks comparison between dogs from rural and urban areas. Is it possible to perform such analysis?

I have three specific comments:

1. There is comparison of age groups in table 1. I'm not sure if I properly understand the results. Which groups were compared? M-W test is used to compare two groups. However, there are 5 age groups. Should the authors use Kruskal-Wallis test?

2. Can the authors discuss in the discussion section the results from lines 192-195? Why the highest PCR prevalence was in the other states of Germany than in the states with the highest seroprevalence? Was the result of older dogs examined in the states with the highest seroprevalence? Or maybe there are other Rickettsia spp. (no Ehrlichia or Analplasma; e.g. R. raoultii) in these states? And the result is caused by crossreaction?

3. All information in supplementary material should be in English.

Author Response

Answer to the comments of reviewer #3

16-January-2023

Dear reviewer #3,

We appreciate the time and effort that you have dedicated to providing valuable feedback on our manuscript. We are grateful for your insightful comments, which have improved the paper. We have incorporated changes that reflect all your comments/suggestions. All changes are highlighted in red in the revised manuscript. Please see below our point-by-point response. If any responses may remain unclear or if you wish additional changes, please do not hesitate to let us know.

Reviewer #3

Comments and Suggestions for Authors

This is interesting and needed epidemiological analysis of canine granulocytic anaplasmosis in Germany. In my opinion the study lacks comparison between dogs from rural and urban areas. Is it possible to perform such analysis?

As a commercial diagnostic laboratory, we were not able to include data on living conditions of the dogs included in the study. We have pointed this out in the limitations section of the manuscript (line 297 ff).

I have three specific comments:

  1. There is comparison of age groups in table 1. I'm not sure if I properly understand the results. Which groups were compared? M-W test is used to compare two groups. However, there are 5 age groups. Should the authors use Kruskal-Wallis test?

The age is coming in five states and the PCR/antibody results in two. None of these variables is ordinal or interval or ratio data. They are all nominal data, and the appropriate test is either one of mid-p exact, chi squared or Fischer’s exact test or the logistic regression. In the opinion of the author’s, there is no need for Kruskal-Wallis testing with such data.

  1. Can the authors discuss in the discussion section the results from lines 192-195? Why the highest PCR prevalence was in the other states of Germany than in the states with the highest seroprevalence? Was the result of older dogs examined in the states with the highest seroprevalence? Or maybe there are other Rickettsia spp. (no Ehrlichia or Analplasma; e.g. R. raoultii) in these states? And the result is caused by crossreaction?

Unfortunately, the discussion is limited due to missing data on tick prevalence in these areas, as indicated in the first part of the discussion. We mentioned the possibilities of cross reactions in antibody testing in the limitations section of the manuscript (lines 308 ff).

  1. All information in supplementary material should be in English.

Has been updated.
